# A Deep Learning Approach for Atrial Fibrillation Classification Using Multi-Feature Time Series Data from ECG and PPG

**DOI:** 10.3390/diagnostics13142442

**Published:** 2023-07-21

**Authors:** Bader Aldughayfiq, Farzeen Ashfaq, N. Z. Jhanjhi, Mamoona Humayun

**Affiliations:** 1Department of Information Systems, College of Computer and Information Sciences, Jouf University, Sakaka 72388, Saudi Arabia; bmaldughayfiq@ju.edu.sa; 2School of Computer Science, SCS, Taylor’s University, Subang Jaya 47500, Malaysia; farzeen.ashfaq@sd.taylors.edu.my (F.A.); noorzaman.jhanjhi@taylors.edu.my (N.Z.J.)

**Keywords:** atrial fibrillation, photoplethysmogram, electrocardiogram, deep learning, BiLSTM network, 1D convolution neural networks

## Abstract

Atrial fibrillation is a prevalent cardiac arrhythmia that poses significant health risks to patients. The use of non-invasive methods for AF detection, such as Electrocardiogram and Photoplethysmogram, has gained attention due to their accessibility and ease of use. However, there are challenges associated with ECG-based AF detection, and the significance of PPG signals in this context has been increasingly recognized. The limitations of ECG and the untapped potential of PPG are taken into account as this work attempts to classify AF and non-AF using PPG time series data and deep learning. In this work, we emploted a hybrid deep neural network comprising of 1D CNN and BiLSTM for the task of AF classification. We addressed the under-researched area of applying deep learning methods to transmissive PPG signals by proposing a novel approach. Our approach involved integrating ECG and PPG signals as multi-featured time series data and training deep learning models for AF classification. Our hybrid 1D CNN and BiLSTM model achieved an accuracy of 95% on test data in identifying atrial fibrillation, showcasing its strong performance and reliable predictive capabilities. Furthermore, we evaluated the performance of our model using additional metrics. The precision of our classification model was measured at 0.88, indicating its ability to accurately identify true positive cases of AF. The recall, or sensitivity, was measured at 0.85, illustrating the model’s capacity to detect a high proportion of actual AF cases. Additionally, the F1 score, which combines both precision and recall, was calculated at 0.84, highlighting the overall effectiveness of our model in classifying AF and non-AF cases.

## 1. Introduction

The heart, as a vital component of the circulatory system, functions as a muscular pump, ensuring the continuous flow of blood to various organs and tissues of the body [1]. The sinoatrial node, frequently referred to as the heart’s natural pacemaker and located in the right atrium, produces electrical impulses that cause the heart to begin pumping. These electrical impulses move through the heart, orchestrating coordinated contractions and promoting the action potential’s quick spread [2].

A heartbeat consists of the contraction (systole) and relaxation (diastole) of the myocardium, regulated by electrical signals originating within the heart and propagated through the cardiac conduction system [3]. The signals begin at the SA node, triggering atrial systole as they travel across the atria, facilitating the transfer of blood to the ventricles. Subsequently, the signals pass through the atrioventricular node, allowing adequate time for atrial blood transfer before continuing towards the ventricles, initiating ventricular systole and pumping blood into the arteries. The relaxation and repolarization of the atria and ventricles follow, denoted as atrial diastole and ventricular diastole, respectively, before the cardiac cycle recommences [3,4]. This contraction and relaxation of the heart muscle, which pumps blood throughout the body, is what is referred to as a heartbeat. In contrast, a heart rhythm is the pattern or sequence of heartbeats over a period of time. The electrical signals that govern how the heart contracts serve as a determinant. Sinus rhythm is the name for the typical heartbeat. However, there can be abnormalities in the heart’s rhythm, known as cardiac arrhythmias, where the electrical signals are disrupted or irregular. Depending on the type of arrhythmia, the heart may beat excessively quickly a condition known as tachycardia or too slowly called bradycardia, or in an irregular pattern [5,6]. These rhythm disturbances can arise from various causes, including structural heart defects, electrolyte imbalances, cardiovascular diseases, and other underlying health conditions [7]. Cardiac arrhythmia encompasses a wide range of abnormal heart rhythms, of which atrial fibrillation is the most common type [8].

Atrial fibrillation affects millions of individuals each year, an estimated 33.5 million people worldwide [9,10]. Its presence increases the risk of strokes, heart failure, and even mortality. According to the Global Burden of Disease study, the worldwide prevalence of AF was reported to be approximately 0.51% of the global population, with an increasing trend [11]. It leads to irregular heartbeats caused by disorganized electrical signals originating from the atria and propagating through the heart. The result is inefficient blood perfusion, leading to symptoms such as fatigue, coagulation issues, strokes, and potential life-threatening consequences [12]. Figure 1 depicts the flow of electrical signals via the cardiac conduction system in atrial fibrillation (right) and normal sinus rhythm (left) [13].

A common test to identify AF is an ECG. It can identify unnatural cardiac rhythms and records the electrical activity of the heart [14]. Electrodes are positioned on the chest, limbs, or both during this painless and non-invasive technique to measure the electrical signals of the heart [15]. Photoplethysmography (PPG) is another non-invasive method that uses light to measure blood volume changes in peripheral blood vessels [16]. When compared to ECG, PPG has a number of benefits. Since it doesn’t require electrodes or close skin contact and it is less complicated. Moreover, wearable gadgets like smartwatches or fitness trackers can incorporate PPG sensors, enabling continuous monitoring throughout the day [17]. PPG is more suited for monitoring during physical exercise because it is less susceptible to motion artefacts. The technique uses light to illuminate the skin and detect the light reflected back to monitor changes in blood volume. The photoplethysmogram is a waveform that is produced when the amount of light reflected changes as a result of the blood pulsing through the vessels. This waveform can reveal details regarding blood flow, heart rate, and perhaps even detect abnormal heartbeats [18].

Machine learning techniques have gained prominence in the analysis and interpretation of physiological signals such as PPG and ECG, enabling advancements in detection, classification, and prediction tasks, with potential applications in various domains [19,20,21,22,23,24,25,26]. By analyzing large datasets of ECG and PPG signals, these models extract meaningful patterns and features associated with AF. Deep learning models, such as convolutional neural networks [27], recurrent neural networks [28] and long short term memory networks [29] also have demonstrated the ability to accurately identify various patterns and features within raw data. These models can be applied to diverse domains, showcasing their efficacy in tasks such as data classification, prediction, and anomaly detection.

Both ML and DL algorithms have also been widely employed in the detection, prediction, and classification of cardiac arrhythmias [30,31], with a particular focus on AF [23]. However, existing studies predominantly utilize ECG datasets for these tasks [32]. Although PPG-based approaches, mainly utilizing smart cameras and transmissive PPG [33], have been explored to some extent, the combined use of PPG and ECG for AF classification remains an underexplored area with limited research conducted thus far.

The aim of this study is to apply deep learning methods in the classification of atrial fibrillation by leveraging multiple data sources, with the objective of achieving higher accuracy. Mainly the study focuses on the comparison of LSTM, BiLSTM, and 1DConv models for the task of atrial fibrillation classification. While the application of deep learning-based methods on transmissive photoplethysmography signals for AF classification is an under-researched area, we propose a novel approach in this study.

The other sections of the study are broken down into the ones below, with Section 2 offering an overview of current research and studies on the identification of atrial fibrillation. The resources and procedures used in this investigation are described in Section 3. The section presents details about the data sources used for the task of AF classification. The results of the investigation are presented in Section 4. The findings are discussed in Section 5. In this section, the results are discussed and compared to past studies while being interpreted in light of the literature that is now accessible. In Section 6, the study discusses the key findings and their implications.

## 2. Literature Review

### 2.1. The Role of ECG in AF Detection

ECG or EKG plays a pivotal role in the detection and diagnosis of AF [34]. The recordings provide valuable insights into the electrical activity of the heart and serve as the gold standard for the diagnosis [35]. By capturing the electrical signals generated during each cardiac cycle, ECG enables the identification of specific patterns and abnormalities associated with AF.

Atrial fibrillation is characterized by the absence of distinct P waves in ECG leads I and III as shown in Figure 2. Instead of the regular and organized P waves observed in normal sinus rhythm, AF exhibits chaotic and irregular atrial electrical activity. These irregular electrical signals in the atria result in an erratic ventricular response, leading to irregular heartbeats.

While ECG is highly valuable for diagnosing AF, it has some limitations [37]. Firstly, AF episodes can be sporadic and transient, making it challenging to capture an episode during a short recording period. Consequently, longer ECG monitoring duration may be necessary to increase the chances of detecting AF episodes. Additionally, ECG requires direct contact with electrodes placed on the body, which can be inconvenient for continuous monitoring in everyday settings [38].

### 2.2. The Significance of PPG in AF Detection

PPG has emerged as a promising non-invasive technique for AF detection. Traditionally used for pulse oximetry and heart rate monitoring, PPG utilizes optical sensors to measure variations in blood volume at the peripheral blood vessels [39]. PPG signals reflect changes in blood flow and can provide valuable information about the cardiovascular system, including the presence of irregular heart rhythms like AF [40]. Figure 3 shows a raw PPG wave segment of a normal sinus rythym and atrial fabrilation patient’s rhythm.

The non-invasive nature of PPG, which may easily be carried out utilising wearable technology or even smartphone applications, is one of its main advantages, according to [42]. This qualifies PPG for continuous monitoring in a variety of contexts, including ambulatory care and home scenarios. Furthermore, PPG is capable of recording physiological data in real-time, enabling the identification of dynamic changes in heart rhythm [43].

### 2.3. Traditional Approach to Classify Atrial Fibrillation

ECG or PPG approaches have typically been used by qualified medical professionals, such as cardiologists, to find AF. To identify cardiac irregularities and disorders, these specialists would examine the ECG waveform patterns, including the size, amplitude, duration, and timing of the various waves, intervals, and segments. The manual interpretation of these waveforms posses several challenges and limitations [44]. Firstly, it relied heavily on the expertise and experience of the interpreting physician, leading to potential variations in diagnoses and the possibility of human error [45]. Moreover, the manual interpretation of ECGs was a labor-intensive task, requiring significant time and effort from the interpreting physician. This could lead to delays in diagnosis, especially in situations where timely intervention was crucial for patient care. Furthermore, the availability of experienced cardiologists for ECG interpretation may be limited, particularly in resource-constrained settings or remote areas [46].

Researchers and developers used computer-assisted image processing and analysis approaches to get over the challenges and constraints [47]. These methods attempted to produce objective and standardised outcomes by automating portions of the ECG interpretation process. To recognise and categorise ECG waveform patterns, computer algorithms were developed, facilitating a quicker and more reliable analysis. Numerous studies investigated techniques for automated ECG interpretation, including template matching, feature extraction, and rule-based algorithms. While computer-assisted interpretation showed promise, it still faced some limitations [48]. Traditional algorithms often relied on handcrafted features, which required expert knowledge and may not capture the full complexity of underlying patterns. They also struggled with handling variability and noise in signals, making them less robust in real-world scenarios. Furthermore, the increasing volume of ECG and PPG data require more advanced and adaptable approaches [49].

### 2.4. Emergence of Machine Learning and Deep Learning

Over the past decade, machine learning and deep learning techniques have emerged as powerful tools for the detection, classification, and prediction of various phenomena, ranging from disease diagnoses in healthcare to fraud detection in finance, and from object recognition in computer vision to sentiment analysis in natural language processing, among many other domains [50,51,52,53,54]. These computational approaches leverage the analysis of large datasets to extract meaningful patterns and features.By training algorithms on labeled datasets, these models have the ability to learn and accurately identify and differentiate between different classes or categories of data in various domains [55,56,57].

#### 2.4.1. Machine Learning Based Atrial Fibrillation Classification

For detection and classification of atrial fibrillation, machine learning algorithms can be trained using a diverse set of features extracted from ECG and PPG signals, including statistical measures [58,59], frequency-domain analysis [60,61], wavelet transformations [62,63,64], and nonlinear dynamics [65]. These features capture specific characteristics of AF, such as irregular heart rate, absence of distinct P waves, and chaotic atrial electrical activity. By feeding these features into supervised learning algorithms such as support vector machines [66], random forests [67], or neural networks [68]. Once trained on big data all these the models have ability to learn and classify new instances as AF or non-AF with high accuracy.

One of the most commonly used models of machine learning in medical research is the Support Vector Machine. SVM has been widely applied to the detection and classification of arrhythmias, with a particular focus on atrial fibrillation. The robustness and accuracy of SVM make it a valuable tool in identifying abnormal heart rhythms [69,70].

Using SVM and the Radial Basis function, [71] focused on the variabilities of electrocardiographic heart rate for AFib detection. The designed model outputs a prediction indicating whether these variabilities in the features belong to the AFib or non-AFib classes. In another study by [72], a Lagrangian SVM was proposed to detect atrial fibrillation using sixteen features as input vector. Other studies which employed AF classification using SVM include [73,74,75,76,77,78].

A different method was used in another study by [79] to identify AF. The analysis of ECG data from portable devices was the study’s main objective. Independent of ECG lead location, a wide range of discriminative features was examined, including features from several domains like time, frequency, time-frequency, phase space, and meta-level. The suggested classification algorithm successfully classified four different ECG types, including AF rhythms, by using a feature selection method based on a random forest classifier. Employing random forest with ecg to detect af was also studied by [80,81,82].

Decision trees are also extensively used in the classification of atrial fibrillation (AF) in various studies. For instance, in a study by [83], decision tree algorithms were employed to classify ECG signals into different rhythm classes, including AF. The study utilized a combination of hand-crafted features extracted from the ECG signals and a decision tree classifier to achieve accurate AF classification. Similarly, in another study conducted by [84], a decision tree-based approach was employed for AF detection using features derived from the heart rate variability analysis.

All the above-discussed studies utilize ECG features for the classification of atrial fibrillation (AF). However, it is worth noting that besides ECG, photoplethysmography (PPG) has also been extensively employed for the classification of heart rhythm, including AF. For example, in a study conducted by [85], PPG signals were utilized along with SVM with linear kernel to classify AF rhythms. The study enables the use of affordable wearable devices with limited processing and data storage resources for long-term ambulatory monitoring of AF. HRV features were extracted in another study by [86] from the inter-beat intervals, and SVM classifier was trained using these features utilizing face video recordings of 200 patients.

#### 2.4.2. Deep Learning Based Atrial Fibrillation Classification

Deep learning, a subset of machine learning, has also shown great potential in AF detection and classification. Deep neural networks can effectively learn complex patterns and dependencies within data without the need for explicit feature engineering [87,88,89]. In case of ECG based time-series data, they can automatically extract hierarchical representations, capturing both local and global patterns indicative of AF. One of the most prominent and extensively used DL models within this domain is the Convolutional Neural Network (CNN). Also, in the case of arrhythmia detection and classification, it has emerged as a widely used and highly effective tool [90,91,92,93].

Using the single lead ECG signal [94] stacked a SVM on statistical features of segment-based recognition units produced by a CNN. The CNN architecture employed automatically classified each segment without the need for feature engineering. For non linear decision boundaries an RBF kernel was utilized with SVM. For the classification of raw ECG time series data [32] employed 1D convolutional neural network with seven FCN layers and achieved 86% validation accuracy. Similarly, one dimensional time series ECG data was utilized by [95] for automatic detection of atrial fibrillation using 1D CNN model. Additionally, a length normalization algorithm was presented to address the challenge of variable-length ECG recordings. Other studies employing 1D CNN for the task include [96,97,98,99,100,101].

Amonst other deep learning models, LSTMs and BiLSTMs have also been widely used in cardiac arrhythmia detection and classification due to their ability to capture temporal dependencies and handle sequential data effectively. These RNN architectures have shown promising results in accurately identifying and categorizing various types of arrhythmias, such as AF, VT, and PVCs. By leveraging the temporal information in ECG signals, LSTMs and BiLSTMs can learn complex patterns and make accurate predictions, improving diagnostic accuracy in clinical settings.

Four different methods to detect AF were employed by [102] with a bidirectional long short-term memory network. The identification and classification of two forms of atrial fibrillation, chronic and paroxysmal, were the study’s main objectives. ECG signal classification has also been done by [103,104]. The first one combined feed-forward and recurrent neural networks to extract relevant features for accurate arrhythmia classification. And the second utilized the LSTM model for the same task. Detection of AF directly by utilizing the raw PPG data, collected over 180 h using multi-sensor wearable devices using LSTM was studied by [105]. The authors reported The area under the receiver operating characteristic (ROC) curve to be 0.9999 for classification labels output every 0.8 s.

Hybrid models combining CNN, LSTM, and BiLSTM architectures also have emerged as popular choices in the literature for the classification of AF, leveraging their ability to capture spatial and temporal dependencies in waveform data effectively. The hybrid CNN-LSTM model studied by [106] is a lightweight 1D deep learning model for ECG beat-wise classification into six classes. In another work proposed by [97] a combination of 1D and 2D CNN is employed. In order to remove PPG data that has been affected by motion artefacts and ambient light interference, the 1D-CNN is used. A 2D-CNN that has been pretrained on ECG data is then utilised to detect AFib. PPIs obtained from wearable technology are used to fine-tune the 2D-CNN. The Table 1 summarizes key studies that have utilized hybrid deep learning models for cardiac arrhythmia classification and related tasks.

### 2.5. Research Findings

Based on the above literature we conclude that one notable research gap in the field of atrial fibrillation detection and classification lies in the exploration of PPG waveform data. While current research predominantly focuses on using machine learning and deep learning with ECG data to classify AF, there has been increasing interest in utilizing PPG for AF detection. However, previous studies on PPG-based AF detection have mainly concentrated on reflective PPG techniques. Reflective PPG measures the light reflected from the skin’s surface and provides information about superficial blood vessels. In contrast, transmissive PPG involves transmitting light through body tissues and measuring the attenuated light after passing through the vessels. Transmissive PPG has the potential to offer deeper insights into the cardiovascular system and may enhance AF detection accuracy. Therefore, the use of bedside PPG data in combination with ECG for AF classification represents a research gap that warrants further investigation.

## 3. Materials and Methods

In this section, we will provide a detailed overview of the dataset used in our study and describe the preprocessing steps undertaken to ensure data quality and consistency. Subsequently, we will delve into the discussion of three deep learning architectures employed in our research: 1DCNN (1D Convolutional Neural Network), LSTM (Long Short-Term Memory), and BiLSTM (Bidirectional Long Short-Term Memory). We will present the architectural details, highlighting the unique characteristics and advantages of each model in the context of our AF classification task. The diagram showing the block diagram of our proposed procedure is shown in Figure 4.

### 3.1. Dataset Preprocessing

The dataset used in this study was extracted from MIMIC PERform dataset [18] which itself is derived from the MIMIC III Waveform Database and offer valuable insights into the physiological state of critically-ill patients. The training dataset contains recordings from 200 patients, with an equal number of adults and neonates, with 100 having NSR and the other half having atrial fibrillation. Likewise, the distribution is the same in the testing dataset. Two instances from the MIMIC PERform dataset are shown in Figure 5, each representing a different physiological signal captured during standard clinical care. The first example shows ECG and PPG data for normal sinus rhythm. While the NSR ECG records the electrical activity of the heart, the NSR PPG visualises the pulsatile variations in blood volume. These signals operate as a benchmark for the common heart activity seen in healthy people.

In contrast, the second example exhibits signals associated with atrial fibrillation. The AFib PPG depicts the irregular and chaotic fluctuations in blood volume caused by the abnormal rhythm, while the AFib ECG highlights the irregularities in the electrical activity of the heart. This example demonstrates the distinct patterns observed in the PPG and ECG signals when the patient is experiencing AFib.

### 3.2. Baseline Architectures

Following a broad introduction to bidirectional LSTM and 1D convolutional neural networks (CNNs), we will explain the architecture used and the rationale behind combining these two types of networks.

#### 3.2.1. 1D Conv Net

CNNs are widely used for image classification tasks, but they can also be applied to time series sequential data using a variant of vanilla CNN called 1D convolution neural network. In 1D CNNs, local patterns or motifs in the sequence are captured by convolutional filters that slide along the temporal axis of the input. Figure 6 shows the architecture of a simple 1D ConvNet.

By applying multiple convolutional layers with increasing filter sizes, CNNs can learn hierarchical representations of the input data, capturing both low-level and high-level features. Such models are useful for extracting meaningful features from sequential data, such as time series, audio signals, and text data.

#### 3.2.2. BiLSTM

Recurrent neural networks of the LSTM variety are particularly good at capturing long-distance dependencies in sequential data. The bidirectional LSTM combines two LSTMs, one of which moves the input sequence forward while the other moves it backward. By processing the input in both ways, they are able to capture dependencies from both past and future context, making them suitable for applications that require a thorough understanding of the sequence. Figure 7 shows the architecture of a vanilla BiLSTM.

### 3.3. Our Proposed Hybrid Architecture

The architecture is designed to classify atrial fibrillation using time-series ECG and PPG data. It consists of the following components. Figure 8 illustrates the model summary.

#### 3.3.1. 1D Convolutional Layers

The initial convolutional layers with increasing filter sizes and ReLU activation functions aim to capture local patterns and features from the ECG and PPG data.The convolutional filters slide along the temporal axis of the input to extract meaningful features.Each convolutional layer is followed by a max pooling layer, which reduces the spatial dimensions and retains important features.

#### 3.3.2. Bidirectional LSTM Layers

The sequence is processed in both forward and reverse directions by the bidirectional LSTM layers, which also collect dependencies and contextual data from both past and future states.By leveraging the bidirectional nature, the model can effectively understand the temporal dynamics of the ECG and PPG signals.

#### 3.3.3. Additional Convolutional and Pooling Layers

Another convolutional layer with a smaller filter size and a max pooling layer follow the bidirectional LSTM layers.This additional layer aims to extract more refined features from the combined information obtained from the previous layers.

#### 3.3.4. Dense Layers and Dropout Layers

Dense layers with leaky ReLU activation functions are used to introduce non-linearity and capture higher-level representations.Dropout layers are included to prevent overfitting by randomly dropping out a fraction of the connections during training.

#### 3.3.5. Output Layer

The binary classification output, indicating the presence or absence of atrial fibrillation, is produced by the final dense layer with a sigmoid activation function.

### 3.4. Reasons for Hybridizing 1D CNN and Bidirectional LSTM

The categorization of atrial fibrillation gains from using both ECG and PPG data. PPG detects variations in blood volume, whereas ECG delivers electrical signals from the heart. The model is able to efficiently process and extract features from both forms of data by combining 1D CNN and bidirectional LSTM. Local patterns and characteristics in the ECG and PPG signals are well-captured by 1D CNNs. The ability to recognise long-term dependencies and comprehend the temporal context is a strength of bidirectional LSTM layers. The capacity of the model to distinguish between regular and atrial fibrillation signals is improved by integrating the two designs, which may capture both spatial and temporal patterns. The hybrid architecture improves performance in AF classification tasks by combining the advantages of bidirectional LSTM with 1D CNN.

### 3.5. Hyperparameter Tuning

To optimize the performance of our models, we conducted parameter tuning experiments. We investigated a number of hyperparameters, such as regularisation methods, dropout rates, and various optimisation approaches. The following are some of the precise specifics of the parameter tweaking experiments:

#### 3.5.1. Regularization

We experimented with different regularization techniques, such as L1 and L2 regularization, to prevent overfitting and improve generalization. We varied the regularization strength to find the optimal balance between model complexity and performance.

#### 3.5.2. Dropout Rate

Dropout is a regularisation strategy that prevents the model from depending too much on particular features by randomly removing a portion of the connections during training. We tested different dropout rates, ranging from 0.1 to 0.5, to find the dropout rate that achieved the best performance.

#### 3.5.3. Optimization Algorithms

We compared different optimization algorithms, such as Adam and SGD to find the algorithm that resulted in faster convergence and better performance.

#### 3.5.4. Final Hyperparameters

By systematically varying these hyperparameters and evaluating the models’ performance using appropriate metrics, we identified the optimal combination of parameters for our AF classification task as presented in Table 2.

These parameter tuning experiments allowed us to fine-tune our models and enhance their performance in accurately classifying atrial fibrillation.

## 4. Results and Experimentation

In the classification of atrial fibrillation using a hybrid 1D-CNN and BiLSTM model, we achieved promising results. The model showed a continuous increase in training and validation accuracy and decrease in the loss over the course of the 25 epochs that made up the training procedure. Details of the evaluation metrics and training procedures are discussed in Appendix A and Appendix B. Figure 9 illustrates the model’s learning process through the training and validation accuracy curves and the loss curves over time. When evaluated on test data the model achieved total loss of 0.1955 and accuracy of 0.9500, which is also highlighted in the each sub-figure. Figure 10 displays the distribution of predicted and actual classes, revealing 50 true negatives (non-AF) and 35 true positives (AF), with no false negatives and no false positives.

To further evaluate the performance of our model, we computed precision, recall, and F1-score for each class. The classification report in Figure 11 provides a comprehensive overview of the model’s performance metrics. Class 0 (non-AF) achieved a precision of 0.77, recall of 1.0, and an F1-score of 0.87. Class 1 (AF) achieved perfect precision, recall, and F1-score, all equal to 1.0.

We also evaluated the model’s sensitivity and specificity as shown in Figure 12. The non-AF category has a sensitivity (recall) of 0.7, which means that 70% of real non-AF cases were accurately identified. The sensitivity for Class AF was 1.0, indicating a 100% accurate identification of AF cases. Class AF’s specificity was 0.7 whereas Class non-AF’s was 1.0, indicating that all non-AF cases were correctly identified.

We also generated the Receiver Operating Characteristic curve as shown in Figure 13 to analyze the model’s overall performance. The area under the ROC curve was calculated as 0.99, indicating excellent discriminative ability between AF and non-AF cases.

Finally, using the same dataset and evaluation metrics, we trained and evaluated a variety of models, including CNN, LSTM, Bidirectional LSTM, BiLSTM, and SVM, to determine how well they performed for classifying atrial fibrillation using PPG and ECG signals as shown Table 3.

In summary, we can conclude that our model successfully identified atrial fibrillation with great accuracy and reliable prediction capabilities.

## 5. Discussion and Findings

In this study, we evaluated the performance of our proposed approach for the classification of atrial fibrillation using a hybrid deep learning model. Our approach incorporated both ECG and PPG signals to improve the accuracy of AF detection.

Comparing our findings with existing literature, we observed that previous studies primarily focused on utilizing only the ECG signal for AF classification. However, our proposed approach went beyond by integrating both ECG and PPG signals. This integration allowed us to capture complementary information from both signals, potentially enhancing the accuracy, precision, recall, and F1-score of the classification model.

Table 4 summarizes the main findings of the comparison between our proposed approach and previous studies in terms of accuracy, precision, recall, and F1-score. It is evident that our proposed approach outperformed the existing studies across all evaluation metrics.

Although the performance of the approach we suggested was greater, it is significant to note that each study’s use of distinct deep learning models. Ref. [118] employed densely connected convolutional neural networks (CNNs) to classify ECG recordings. The model consisted of a main CNN that processed 15-s ECG segments and a secondary CNN that analyzed shorter 9-s segments. In [119], a deep classifier was constructed using a combination of CNNs and LSTM units. The network architecture included pooling, dropout, and normalization techniques to improve accuracy. An ensemble classifier was then created by cross-validating ten standalone models based on the same architecture. For [120], the researchers explored the application of machine learning and deep learning algorithms for AF detection. Specifically, they compared the performance of LSTM and CNN algorithms against traditional machine learning classifiers such as support vectors and logistic regression. The results of their experimentation revealed interesting findings. The researchers reported that a simple CNN model achieved an accuracy of 0.865, indicating its effectiveness in distinguishing AF cases from normal sinus rhythm. They also evaluated a CNN+LSTM hybrid model, which combined the strengths of both architectures, resulting in an accuracy of 0.811. Additionally, they explored the performance of a residual network (ResNet) model, which achieved an accuracy of 0.792. This finding suggests that the ResNet architecture may not have been as effective in capturing the relevant features for AF detection in this particular study. Furthermore, the researchers investigated the performance of a standalone LSTM model, which achieved an accuracy of 0.875. This demonstrates the potential of LSTM networks in capturing the temporal dependencies present in the ECG signals. Building upon this we extended thier findings with hybrid CNN and BiLSTM network achieving an impressive accuracy of 0.95, surpassing the performance of the individual models and demonstrating the effectiveness of this combined approach for AF detection. These findings are summarized in Table 5.

The following contributions were made as a result of our method integrating both ECG and PPG signals as multi-featured time series data for deep learning models.

Our study utilized both ECG and PPG signals for AF classification. By leveraging the complementary information provided by these two modalities, our research aims to improve the accuracy of AF detection.Our methodology made use of multi-feature time series data from ECG and PPG signals rather than depending just on a single feature. This thorough representation enables the collection of various patterns and AF-related traits, improving classification performance.With the use of a hybrid model that combines a 1D Convolutional Neural Network with a Bidirectional Long Short-Term Memory architecture, we proposed a unique method for classifying atrial fibrillation. Due to the deep learning models’ integration, we were able to accurately detect temporal dependencies in the data.Through trials and evaluation, we were able to classify AF and non-AF cases with a high degree of accuracy of 95%. The usefulness of your suggested technique was also highlighted by the great performance of other evaluation criteria like specificity and sensitivity.

## 6. Conclusions

For the categorization of atrial fibrillation, we used a hybrid model in this study that combines a simple 1D CNN with a BiLSTM. The use of BiLSTM is an important component of our research because it has been comparatively underutilised in prior AF detection investigations. Moreover, the utilization of a simple 1D CNN as the base model is advantageous in terms of its lightweight architecture, which allows for efficient computation and reduced model complexity. This is particularly important in the context of AF detection, where real-time processing and low computational requirements are crucial for practical applications. In addition to this one other significance of our approach lies in the incorporation of PPG alongside ECG, which allowed us to leverage the complementary information from both signals. By considering additional features provided by PPG, we improved the detection and classification of AF, enhancing the overall performance of the model. Although our suggested solution outperformed existing ones in terms of performance, it is crucial to confirm its efficacy on a bigger and more varied dataset. This would guarantee our model’s resilience and generalizability across various patient groups and data collection environments. In future we plan to the deployment and evaluation of our proposed model in a real-world clinical setting would provide valuable insights into its practical feasibility and performance in a clinical context. Such studies can contribute to the translation of our research into clinical practice, ultimately benefiting patients by enabling early and accurate detection of atrial fibrillation.

## Figures and Tables

**Figure 1 diagnostics-13-02442-f001:**
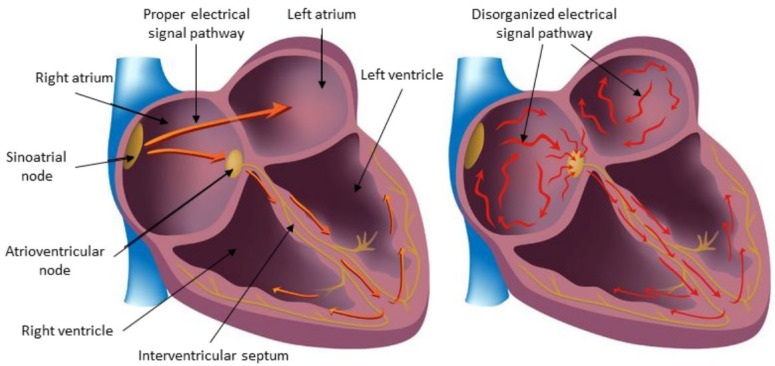
Electrical signal Flow in Atrial Fibrillation (**right**) and Normal Sinus Rhythm (**left**) [13].

**Figure 2 diagnostics-13-02442-f002:**
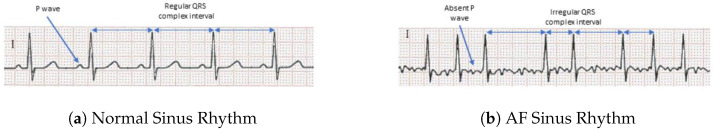
Lead I ECG recording. (**a**) ECG Recording of NSR and (**b**) ECG Recording of AF. In NSR, a clear and distinct P wave is observable, which is absent in AF. Additionally, the interval between consecutive QRS complexes remains constant in NSR, whereas it becomes irregular in AF. Adapted from [36].

**Figure 3 diagnostics-13-02442-f003:**
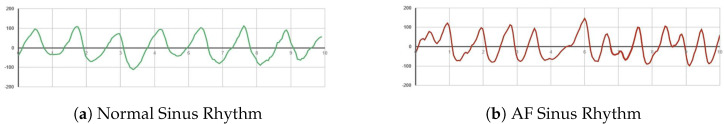
Raw PPG waveform of (**a**) Normal Sinus Rhythm (**b**) Atrial Fibrillation. As can be seen that the AF waveform exhibits the distinctive pattern of being “irregularly irregular”, characterized by variations in both the amplitude and period of the waveform [41].

**Figure 4 diagnostics-13-02442-f004:**
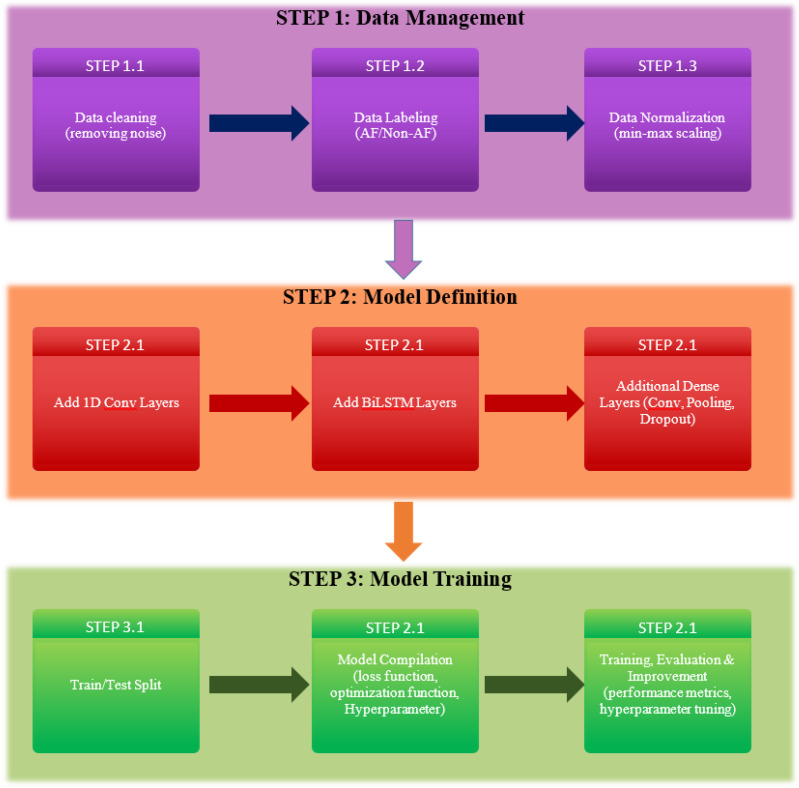
Block Diagram of our proposed procedure.

**Figure 5 diagnostics-13-02442-f005:**
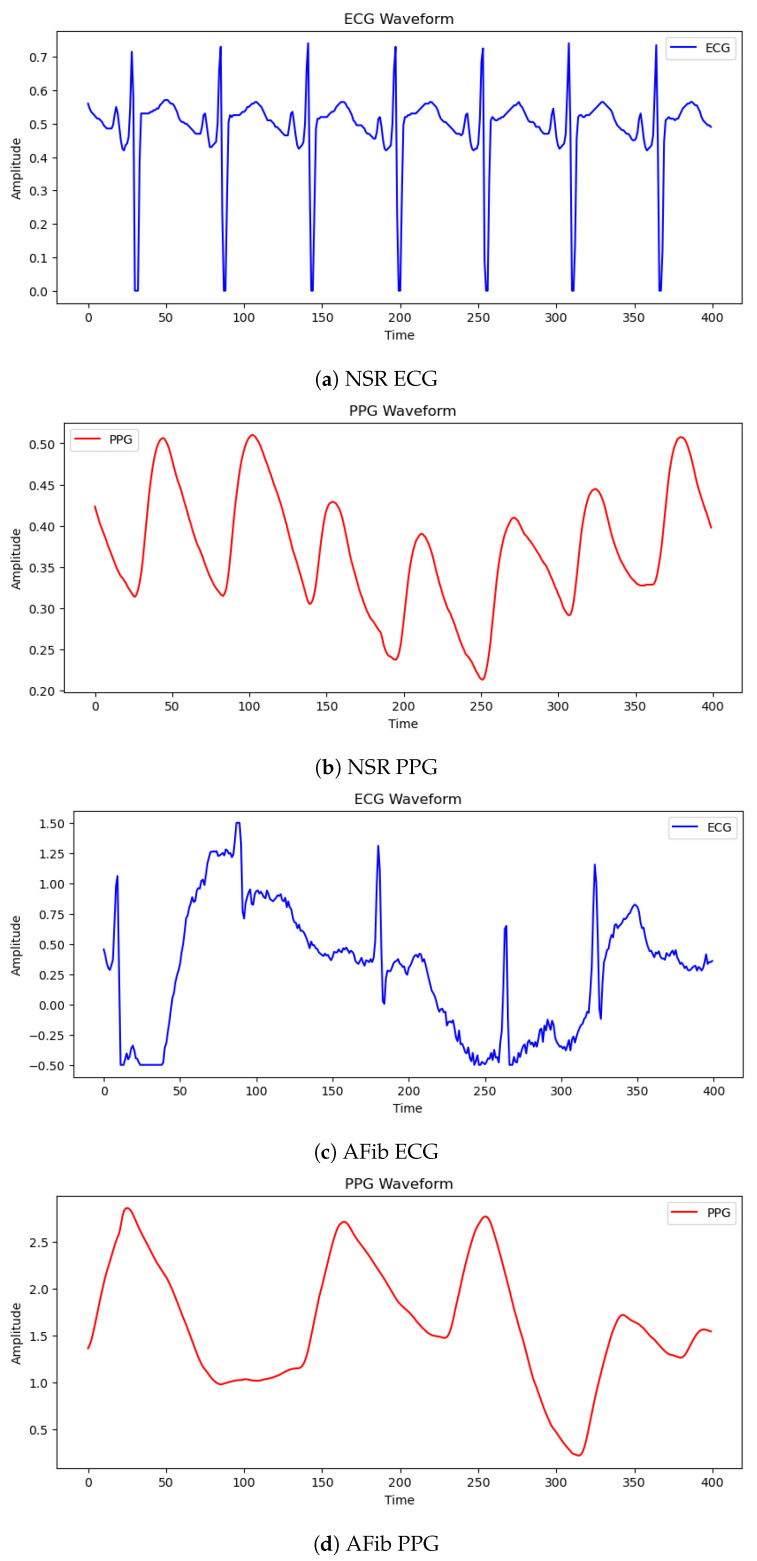
Physiological signals from the MIMIC PERform dataset. Subfigure (**a**) shows NSR ECG, (**b**) displays NSR PPG, (**c**) depicts AFib ECG, and (**d**) exhibits AFib PPG.

**Figure 6 diagnostics-13-02442-f006:**
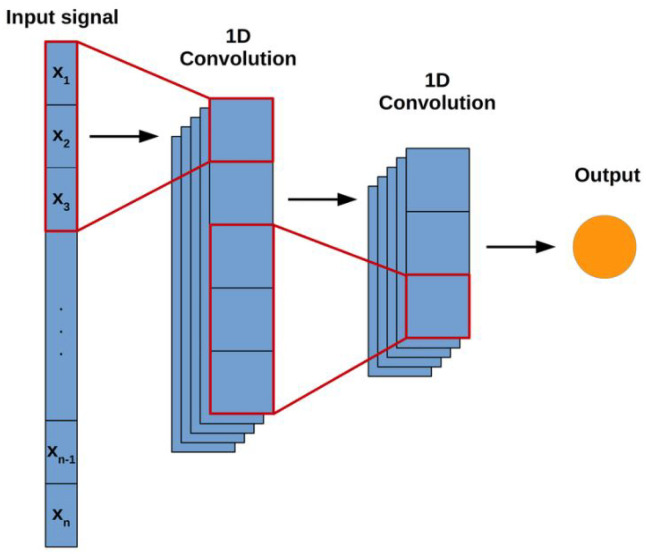
The architecture of a Simple 1D CNN adapted from [116].

**Figure 7 diagnostics-13-02442-f007:**
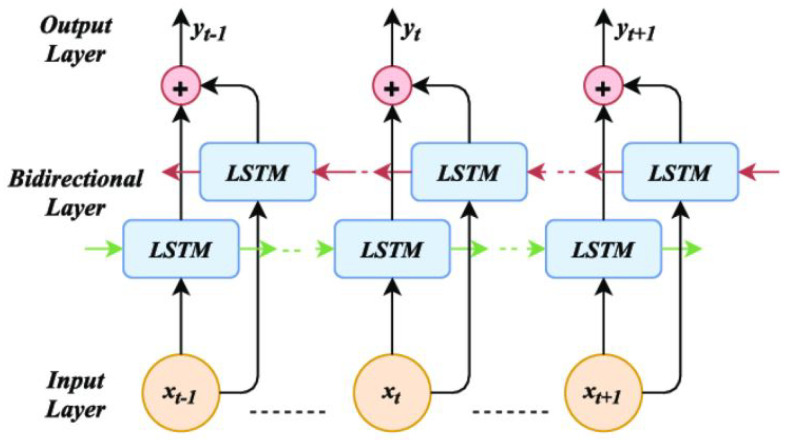
The architecture of a Simple BiLSTM network adapted from [117].

**Figure 8 diagnostics-13-02442-f008:**
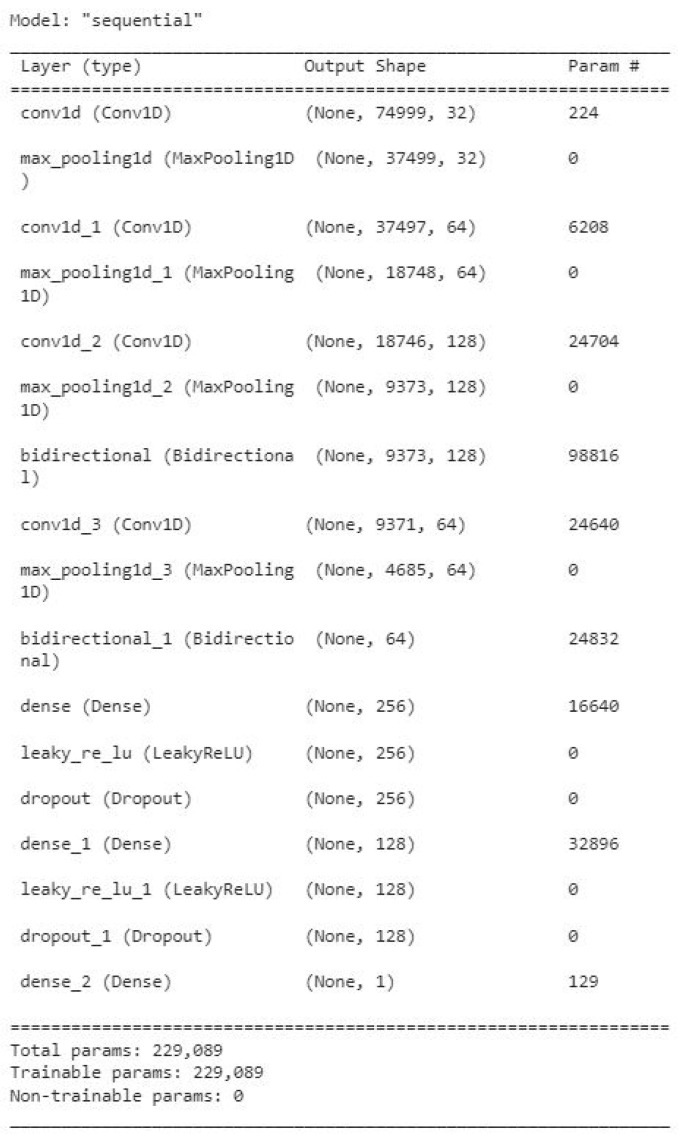
The summary of our proposed architecture.

**Figure 9 diagnostics-13-02442-f009:**
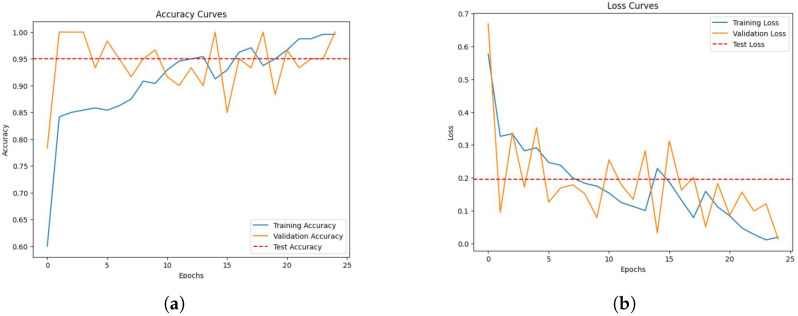
Training and validation accuracy and loss curves during the model training process. The testing accuracy and loss are also displayed. (**a**) Training and Validation Accuracy Curve along with Testing Accuracy, (**b**) Training and Validation Loss Curve along with Testing Loss.

**Figure 10 diagnostics-13-02442-f010:**
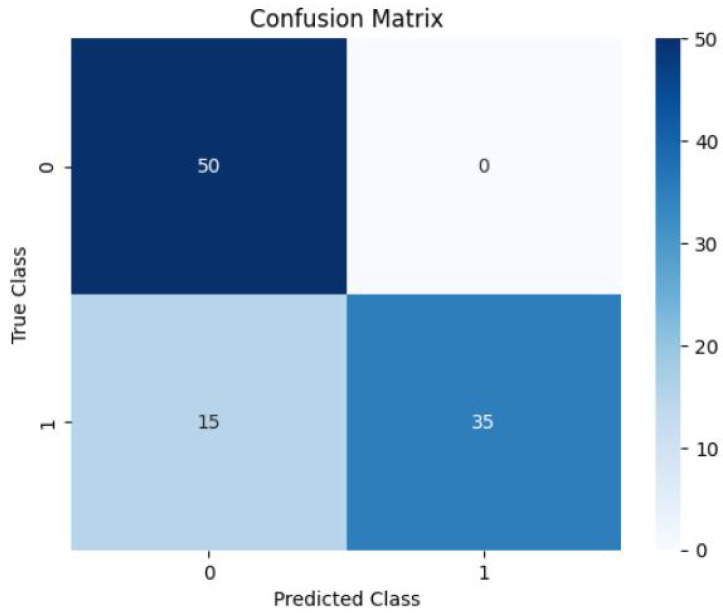
Obtained Confusion Matrix.

**Figure 11 diagnostics-13-02442-f011:**
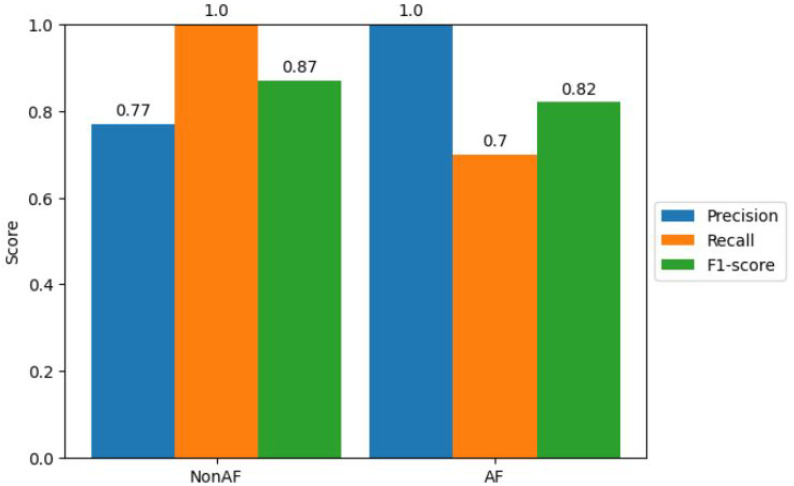
Precision, Recall, and F1-score for Atrial Fibrillation Classification.

**Figure 12 diagnostics-13-02442-f012:**
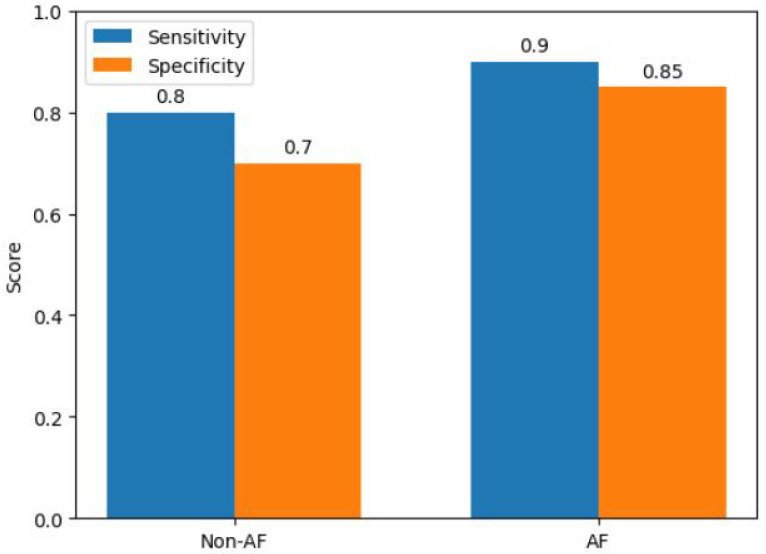
Sensitivity and Specificity for Atrial Fibrillation Classification—Comparison of class-wise sensitivity and specificity values for class Non-AF and class AF. The bar plot illustrates the performance of the classification model in terms of correctly identifying positive (sensitivity) and negative (specificity) instances for each class.

**Figure 13 diagnostics-13-02442-f013:**
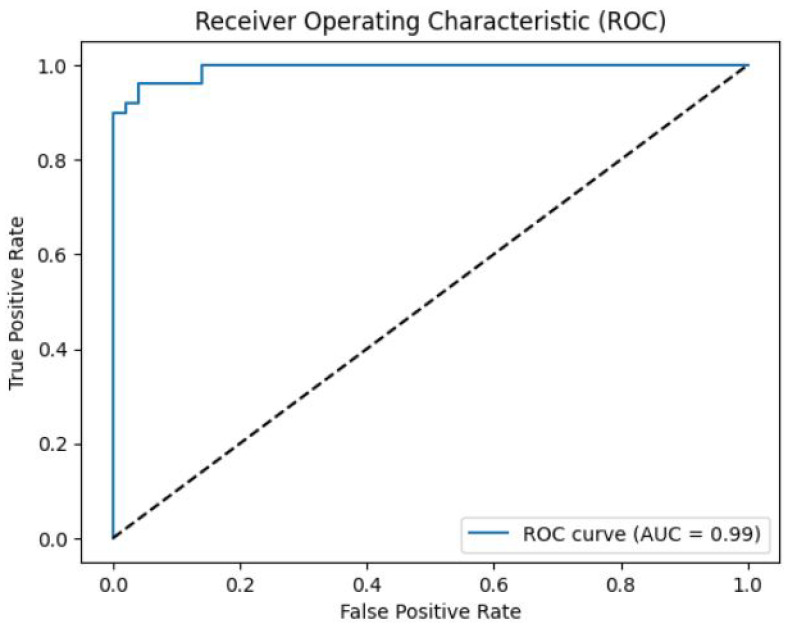
The ROC curve illustrates the trade-off between the true positive rate (sensitivity) and the false positive rate for the atrial fibrillation classification model. The AUC is a measure of the model’s overall performance, with higher values indicating better discrimination between the classes. The diagonal line represents the performance of a random classifier. The AUC value for this model is 0.85, indicating good predictive performance.

**Table 1 diagnostics-13-02442-t001:** Studies Utilizing Hybrid Deep Learning Models for Cardiac Arrhythmia Classification and Related Tasks.

Study	LSTM	BiLSTM	CNN	Other	ECG	PPG	Task	Evaluation Metrics	Dataset
[102]	🗸	🗸	🗸		🗸		AF Detection	Training Accuracy, Test Accuracy	4th China Physiological Signal Challenge—2021
[107]		🗸		KMMBO	🗸		Arrhythmia Classification	Accuracy, Sensitivity	China Physiological Signal Challenge 2018, MIT-BIH
[108]	🗸	🗸		Attention Mechanism			Heart Rate Prediction	RMSE	-
[109]	🗸		🗸		🗸		Arrhythmia Classification	Accuracy, AUC-ROC	MIT-BIH Arrhythmia Database, AFDB, NSRDB
[110]	🗸		🗸		🗸		AF Detection	Weighted F1 Score	MIT-BIT Arrhythmia Physionet
[111]	🗸			Non Hybrid		🗸	Blood Pressure Prediction	Mean Error, Standard Deviation	MIMIC-III Waveform Database
[112]				SVM	🗸	🗸	AF Classification	Accuracy	MIT-BIH Atrial Fibrillation Database, 2017 Challenge Database
[113]				Random Forest	🗸	🗸	AF Detection	Accuracy	ECG Databases, PPG Databases
[114]	🗸		🗸		🗸	🗸	AF Detection	F1 value, Overall Accuracy	ECG Databases, PPG Databases
[115]			🗸		🗸	🗸	AF Classification	Accuracy, Sensitivity, Specificity	MIT-BIH ECG Databases, Unseen PPG Dataset

**Table 2 diagnostics-13-02442-t002:** Final Hyperparameters.

Hyperparameter	Value
Conv1D Filters	32, 64, 128
Conv1D Kernel Size	3
LSTM Units	64, 32
Dense Units	256, 128
Dropout Rate	0.5
Regularizer	L2
Optimizer	Adam
Learning Rate	0.001
Epochs	25
Batch Size	20
Validation Split	0.2

**Table 3 diagnostics-13-02442-t003:** Comparison of precision and accuracy for different ML and DL methods on same dataset.

Method	Precision	Accuracy
Proposed (Hybrid)	0.85	0.95
CNN	0.78	0.90
LSTM	0.75	0.88
BiLSTM	0.82	0.92
SVM	0.70	0.85

**Table 4 diagnostics-13-02442-t004:** Comparison of results with benchmark studies.

Study	Metrics			
	Accuracy	Precision	Recall	F1
[118]	80.0	0.80	0.79	0.80
[119]	77.1	0.77	0.76	0.76
[120]	86.5	0.86	0.85	0.86
Our proposed	95.0	0.88	0.85	0.84

**Table 5 diagnostics-13-02442-t005:** Performance of deep learning models in classification of AF.

Model	Accuracy
Simple CNN	0.865
CNN+LSTM	0.811
ResNet	0.792
LSTM	0.875
1DCNN-BiLSTM (Proposed)	0.950

## Data Availability

Not applicable.

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
