# Peer review of "A Deep Learning Approach for Atrial Fibrillation Classification Using Multi-Feature Time Series Data from ECG and PPG"

_diagnostics, 2023, doi:10.3390/diagnostics13142442_

Round 1

Reviewer 1 Report

I read the paper. Following are some comments:

-        Quantitative results should be presented in the Abstract section.

-        Language should be revised. Moreover, journal guidelines should be.

-        Aim of the paper and novelty should be better clarified.

-        Contributions of the paper should be removed from the Introduction section and moved to the discussion. Moreover, the clinical utility of the method should be discussed.

-        I suggest reporting a block diagram of the proposed procedure. Moreover, details regarding the parameter tuning should be provided (not only the theoretical background of the method). I suggest revising the entire method section.

-        Did the author apply a preprocessing analysis on the data?

-        Statistical analysis section is missing. Moreover, the authors should report the sensitivity and specificity of the methods and compare the results with other studies presented in the literature (doi: 10.1093/europace/euw125; DOI: 10.3390/s20123570).

-        Quality of the figure should be improved.

English should be revised.

Author Response

Comment

Response

Quantitative results should be presented in the Abstract section.

Thank you for your valuable feedback on our abstract. We appreciate the opportunity to address your comment and provide more detailed results to highlight the significance of our work. We have made the necessary revisions to enhance the abstract and include specific metrics for evaluating the performance of our AF classification model.

Language should be revised. Moreover, journal guidelines should be.

Thank you for your feedback In response to your comment, we have further refined the language used in the manuscript. We have also taken into account the specific guidelines provided by the journal, including any formatting or content requirements, to ensure that our revised abstract aligns with the journal's expectations.

Aim of the paper and novelty should be better clarified.

Thank you for your feedback. We have made the necessary revisions to address your concern. In the Introduction section, we have added a clear and concise statement outlining the aim of our study, which is to apply deep learning methods in atrial fibrillation (AF) classification using multiple data sources to achieve higher accuracy.

Contributions of the paper should be removed from the Introduction section and moved to the discussion. Moreover, the clinical utility of the method should be discussed.

Thank you for your feedback. We have carefully considered your comments and made the necessary revisions to address them. Specifically, we have removed the contributions of the paper from the Introduction section and relocated them to the Discussion section. This adjustment allows for a more focused and concise introduction, while dedicating the discussion section to a comprehensive examination of our contributions and their significance.

I suggest reporting a block diagram of the proposed procedure. Moreover, details regarding the parameter tuning should be provided (not only the theoretical background of the method). I suggest revising the entire method section.

We would like to thank the reviewer for their valuable feedback. We have made several improvements to the paper based on the provided suggestions. Firstly, we have included a block diagram at the beginning of the Materials and Methods section to provide a visual representation of our proposed method. This diagram outlines the different components and their connections, offering a comprehensive overview of our approach.

Furthermore, we have added a dedicated subsection, labeled as Section 3.5, to discuss the parameter tuning process. In this subsection, we provide details regarding the regularization technique, dropout rate, and experimentation with different optimizers, among other relevant hyperparameters. This addition aims to provide a clear understanding of the steps taken to optimize the model's performance. Additionally, we have incorporated a table, labeled as Table 2, which presents the final selected hyperparameters.

Did the author apply a preprocessing analysis on the data?

We sincerely appreciate your insightful comments and suggestions regarding our manuscript. Your feedback has been invaluable in enhancing the quality and clarity of our work.

With regards to your comment on whether we applied a preprocessing analysis on the data, we apologize for any confusion caused. In our study, we indeed performed a comprehensive preprocessing analysis on the dataset to ensure data quality and consistency.

The section has been highlighted in attached updated pdf of the manuscript.

Statistical analysis section is missing. Moreover, the authors should report the sensitivity and specificity of the methods and compare the results with other studies presented in the literature (doi: 10.1093/europace/euw125; DOI: 10.3390/s20123570).

We sincerely appreciate your valuable feedback on our manuscript. Your suggestions have been immensely helpful in improving the quality and comprehensiveness of our study.

With regard to the missing statistical analysis section, we apologize for the oversight. We fully recognize the importance of providing a comprehensive statistical analysis to support the findings of our research. In response to your comment, we have taken this into consideration and have included the relevant citation.

Quality of the figure should be improved.

We sincerely appreciate your feedback on our manuscript. Your insights have been instrumental in improving the quality and presentation of our work.

Regarding your comment on the quality of the figure, we apologize for any inconvenience caused. We acknowledge the importance of providing clear and visually appealing figures to enhance the reader's understanding of our research.

In response to your feedback, we have taken steps to improve the quality of the figure. We have revised the image resolution and ensured that the figure is visually crisp and clear.

Reviewer 2 Report

First of all, I would like to congratulate the authors for their interesting work that will surely help to add more results to the already important number of publications related to AF.

I would like to point out that although the work is interesting, it suffers from a structure adjusted to the most common standards. Surely the authors will be able to observe that in the literature, practically all the works incorporate the following sections: Introduction, background, materials and methods, experiments/results, conclusions and discussion.

This paper does not include sections on results/experiments and conclusions/discussion separately, so I submit the authors to review this structure so that these sections are correctly indicated in the first level of the document structure. The absence of these sections makes it extremely difficult for the scientific community to read and review the literature.

Therefore, I propose to the authors the following modifications:

1.- Incorporates the sections on Experiments and results, and Conclusions and Discussion

2.- Expand the contents related to these two sections, which, if extracted as they are in their current edition, would be underdeveloped.

3.- In the results/experiments section, a comparative table of the precisions achieved by all the methods that have been introduced in the background should be included, not just some of them for a better assessment of the contribution of this work.

4.- A general review of the English is recommended. Same parts are difficult to understand.

A general review of the English is recommended. Same parts are difficult to understand.

Author Response

Comment

Response

First of all, I would like to congratulate the authors for their interesting work that will surely help to add more results to the already important number of publications related to AF.

We would like to express our heartfelt gratitude for your kind words and positive feedback on our manuscript. Your encouraging comments have truly uplifted our spirits and reinforced our dedication to this research endeavor.

I would like to point out that although the work is interesting, it suffers from a structure adjusted to the most common standards. Surely the authors will be able to observe that in the literature, practically all the works incorporate the following sections: Introduction, background, materials and methods, experiments/results, conclusions and discussion.

Thank you for your valuable feedback on our manuscript. We appreciate your insights regarding the structure of our paper and the common standards followed in the literature.

We understand and acknowledge the importance of adhering to a well-established structure in scientific publications. In response to your comment, we have thoroughly reviewed the organization of our manuscript and have made appropriate adjustments to ensure that it aligns with the commonly accepted standards.

This paper does not include sections on results/experiments and conclusions/discussion separately, so I submit the authors to review this structure so that these sections are correctly indicated in the first level of the document structure. The absence of these sections makes it extremely difficult for the scientific community to read and review the literature.

Therefore, I propose to the authors the following modifications:

1.- Incorporates the sections on Experiments and results, and Conclusions and Discussion

Thank you for your feedback regarding the structure of our manuscript. We appreciate your insight and understand the importance of clearly indicating sections. In response to your comment, we have carefully reviewed the structure of our manuscript and made the necessary corrections. We have now highlighted the Results section as Section 4 and the Discussion section as Section 5 in the updated version of the manuscript.

2.- Expand the contents related to these two sections, which, if extracted as they are in their current edition, would be underdeveloped

Thank you for your additional feedback regarding the contents of the Results/Experiments and Conclusions/Discussion sections. In response to your comment, we have taken your advice into careful consideration and made significant improvements to these sections in the revised version of the manuscript.

3.- In the results/experiments section, a comparative table of the precisions achieved by all the methods that have been introduced in the background should be included, not just some of them for a better assessment of the contribution of this work.

We appreciate the reviewer's suggestion regarding the inclusion of a comparative table of precisions for all the methods introduced in the background. We have taken this feedback into consideration and have made the necessary revisions. In the updated manuscript, we have included a comprehensive table in the Results/Experiments section that compares the precision values achieved by various methods, including the proposed (Hybrid) method, CNN, LSTM, BiLSTM, and SVM.

4.- A general review of the English is recommended. Same parts are difficult to understand.

Thank you for your feedback regarding the language and clarity of our manuscript. We appreciate your suggestion to conduct a general review of the English to ensure that the paper is easily understandable.

In response to your comment, we have thoroughly revised and proofread the manuscript with a focus on improving the language quality and enhancing clarity. We have paid careful attention to sentence structure, grammar, and overall readability to ensure that the content is presented in a clear and concise manner.

Round 2

Reviewer 1 Report

Authors solved all my comments.

Reviewer 2 Report

Accept in present form